# The Burden of Spinal Muscular Atrophy on Informal Caregivers

**DOI:** 10.3390/ijerph17238989

**Published:** 2020-12-02

**Authors:** Isaac Aranda-Reneo, Luz María Peña-Longobardo, Juan Oliva-Moreno, Svenja Litzkendorf, Isabelle Durand-Zaleski, Eduardo F. Tizzano, Julio López-Bastida

**Affiliations:** 1Department of Economic Analysis and Finances, Faculty of Social Sciences, University of Castilla-La Mancha, 45600 Talavera de la Reina, Spain; 2Department of Economic Analysis and Finances, Faculty of Law and Social Sciences, University of Castilla-La Mancha, 45071 Toledo, Spain; luzmaria.pena@uclm.es (L.M.P.-L.); juan.olivamoreno@uclm.es (J.O.-M.); 3Health Economics Department, Center for Health Economics Research Hannover, 30159 Hannover, Germany; sl@cherh.de; 4Department of Public Health, Hospital Albert Chenevier, 94000 Paris, France; isabelle.durand-zaleski-ext@aphp.fr; 5Department of Clinical and Molecular Genetics Hospital Vall d’Hebron, and CIBERER, 08035 Barcelona, Spain; etizzano@vhebron.net; 6Medicine Genetics Group, Vall d’Hebron Research Institute (VHIR), 08035 Barcelona, Spain; 7Department of Nursing, Faculty of Health Sciences, University of Castilla-La Mancha, 45600 Talavera de la Reina, Spain; julio.lopezbastida@uclm.es

**Keywords:** spinal muscular atrophy, informal care, burden of disease

## Abstract

Spinal muscular atrophy (SMA) is one of the most common severe hereditary diseases of infancy and early childhood. The progression of this illness causes a high degree of disability; hence, a significant burden is experienced by individuals with this disease and their families. We analyzed the time taken to care for patients suffering from SMA in European countries and the burden on their informal caregivers. We designed a cross-sectional study recording data from France, Germany, Spain and the United Kingdom. The primary caregivers completed a self-administered questionnaire that included questions about the time of care, The Zarit Burden Interview, type of SMA and socio-demographic characteristics. Multivariate analyses were used to study the associations between the type of SMA, time of care and burden supported by informal caregivers. The caregivers provided 10.0 h (SD = 6.7) per day of care (the principal caregivers provided 6.9 h, SD = 4.6). The informal caregivers of patients with type I SMA had a 36.3 point higher likelihood (*p* < 0.05) of providing more than 10 h of care per day in comparison with caregivers of patients with type III SMA. The severity of the disease was associated with more time of care and a higher burden on the caregivers.

## 1. Introduction

Spinal muscular atrophy (SMA) is the second most common severe hereditary disease of infancy and early childhood, with an estimated incidence of 1/5000 to 1/10,000 births and a carrier frequency between 1/35 and 1/50 [1].

The progression of this illness causes muscular atrophies, respiratory failure and death [2,3]. Individuals with SMA have a high degree of morbidity and mortality, particularly those with early disease onset [4]. Type I SMA (or Werdnig–Hoffmann disease) is the most severe form. It is characterized by generalized muscle weakness and hypotonia at birth or within the first six months of life. Children have limited head control, are never able to sit unaided and have swallowing and respiratory problems. According to natural history studies [2,3], the vast majority of type I patients die within the first two years of life, and survival may be possible with adequate ventilation support [5]. Type II SMA (the intermediate form) is distinguished by the age of onset, which usually occurs after 6 months and before 18 months of life. Children are able to sit, but they will never walk unaided and are wheelchair-bound their whole life. Scoliosis, nutritional problems and contractures are common complications that need proper intervention. Type III SMA (or Kugelberg–Welander disease) is the least severe and starts after 18 months. Patients are able to walk, but they may lose this ability later in life. Furthermore, due to the considerable disability that SMA causes, most of the patients are unable to perform their Activities of Daily Living (ADL) and rely on family and/or social services for support. Thus, a significant burden falls on individuals with SMA and their families [6].

However, although the role played by the family has been an unseen social resource until recently [7], there is an increasing amount of literature that has tried to assess the duration of care provided [8] in other non-rare diseases. Indeed, it has been proven that the effect of the inclusion of informal care costs, not only in cost-of-illness studies [9] but also in economic evaluations [10,11], may have an important impact on the results. However, the extent of informal care for patients with SMA has only been assessed in two studies [12,13]. Other studies have focused on healthcare costs (omitting a wider perspective from the analysis, which includes non-healthcare costs) [14,15,16] or qualitatively analyzed the needs of these patients [17]. No previous studies have focused on how much care time is needed by patients with SMA or what informal care actually entails for this disease in different European countries using prospective transversal data. In most high-income countries, long-term care schemes are evolving towards shared responsibility models [18,19], especially in such disabling illnesses as SMA. These models presuppose that society assumes neither the families nor the government should take 100% of the responsibility for the care provided to people with limited autonomy and that the financial and personal burdens should be shared. In this sense, assessing the situation of carers (in terms of the burden borne) has become increasingly relevant in order to address the design and application of policies that combine a rational and efficient use of resources while remaining equitable.

When assessing the economic impact of SMA, taking into account the high healthcare cost, the cost of informal care has been revealed as the most important factor to consider [12,13], reaching €21,127 in Spain [13], €20,170 in Germany [12], €40,526 in the UK and €25,619 in France [20]. The percentage of total cost-of-illness is close to 80% in some countries. In Germany, the cost of informal care has been estimated as 77% of total cost-of-illness, while it was estimated as 75% in the UK and 80% in France [20]. This implies that a large amount of non-professional (or informal) resources must be mobilized to provide the care that patients with SMA need. However, there is a great lack of knowledge about aspects related to the duration of care and the specific tasks provided, as well as the burden that caregivers must bear due to the high dependence that this disease generates.

The main aim of the study was to analyze the burden taken on by informal caregivers (non-professional) of patients with SMA. Firstly, we analyzed the distribution of caregiving time provided depending on the task to present an overview of the patient’s needs. Secondly, we analyzed the main factors associated with the caregiving time and the burden supported by caregivers.

## 2. Materials and Methods

This study was a cross-sectional analysis that involved the families of individuals diagnosed with SMA who received outpatient care and were living in the community. Four European countries participated in the data collection: Spain, Germany, the United Kingdom (UK) and France. The fieldwork was carried out between July 2015 and November 2015, and the questionnaires were distributed by e-mail through patient organizations. People eligible to participate in this study were children/adolescents diagnosed with SMA and their main caregivers. Two different questionnaires were filled in: one giving information about patients, and the other about primary caregivers. All patients/caregivers were informed about the study’s objectives and about data confidentiality and were asked to confirm their understanding of the study conditions and their agreement to participate. The study was approved by the Medizinische Hochschule Hannover Ethics Committee (Nr 2711-2015).

The data were obtained from a questionnaire completed by primary caregivers through a website specially developed for this study. The questionnaire included questions used previously in the BURQOL (Social Economic Burden and Health-Related Quality of Life of Patients with Rare Diseases in Europe) project [21], especially those related to the consumption of healthcare resources. Questions were adapted to the SMA illness by an expert committee, whose members included physicians and health economics researchers. The questionnaires were first sent to patients’ associations to obtain feedback about the length of the questionnaires and the rationale of the questions included or suggested by the expert committee. The questionnaires included sociodemographic questions as well as information about the type of SMA on the basis of age at onset and clinical severity of the disease according to the International SMA Consortium [22,23]. We defined an informal caregiver as any relative, friend or other individual who informally cared for a person affected by SMA [24]. Informal or family care is usually provided by one or more members in the social environment of the person who needs such care, mainly by the parents or other members of the immediate family. Informal caregivers are not subject to labor rights or obligations. This implies the absence of a regulated daily/weekly schedule or planned holidays. Also, informal caregivers do not receive any remuneration for their services, although there may be specific benefits, training programs and support for caregivers in some countries. The care activities provided by informal caregivers entail all kinds of activities related to the disease that the care recipient cannot perform alone due to the lack of personal autonomy because of the illness. This definition is widely used in the literature [25,26,27,28,29]. We used two questions to identify informal care: the first question asked whether the patients needed assistance due to the illness (does the patient need a carer to help with his/her daily activities (toilet, washing, dressing, helping him/her move, drug treatments, etc.) because of SMA?). The second question asked who provided the assistance. The patients could answer: “a family member”, “another unpaid person” or “a professional carer”. We only included the patients who answered with the first two options in the analysis of the burden of SMA on informal caregivers. Later, in the questions regarding the duration of care, we asked the main informal caregiver how much time was spent on caregiving by people other than the main caregiver. In this sense, we considered the primary caregiver as the person who had the leading role and responsibility in the performance of the caregiving tasks, provided the greatest amount of care and bore most of the burden. The other caregivers were those who complimented the care provided by the primary ones.

The questionnaires for carers included socioeconomic questions, as well as the Zarit Burden Interview [30]. Using this structure (22-item version), the caregiver responded on the Likert scale, with options ranging from 0 (never) to 4 (nearly always). The total scores ranged from 0 to 88, and scores below 21 points corresponded to low or no burden, while scores above 61 points corresponded to a severe burden. Moreover, specific questions were also included to estimate the amount of care time spent in a typical day on help with Basic Activities of Daily Living (BADL), such as basic hygiene activities, dressing or changing the patient, bathing or showering, feeding and helping the patient to move. We also asked how much time they spend each month on Instrumental Activities of Daily Living (IADL), such as preparing special meals or administering drugs, the time spent on taking the patient for medical visits, medical tests, traveling, attending to financial or administrative matters and social/leisure activities.

We used the recall method [31] for every BADL and IADL question to reveal the duration of care provided by the caregivers. In studies where children are included, estimating the duration of care provided by caregivers due to the illness is a challenge as it is difficult for the caregiver to distinguish how much time is due to the illness and how much is concerned with the progress of the child’s development. For this reason, the questions about the duration of care were formulated as: “how much time do you spend on each of the activities related to the patient’s disease?”. Moreover, “to the patient’s disease?” was underlined and highlighted to ensure that the caregiver paid attention to the importance of this point. Finally, to avoid overestimation, the caregiving time provided was conservatively capped at a maximum of 16 h per day.

### Statistical Analysis

The purpose of the statistical analysis was to show the effect of the severity of the illness on informal caregivers. We used the SMA classification forms (type I, type II and type III) as a proxy for the severity of the illness. We chose two variables to assess the burden of SMA on informal caregivers: (i) the duration of care provided each day, and (ii) the Zarit Burden Interview score. Due to the asymmetric distribution of the data on the duration of caregiving, we categorized the hours of informal care into three groups: (i) less than 5 h per day; (ii) between 5 and 10 h per day and (iii) more than 10 h of care in a day. This classification allowed us to capture the ordinal nature of the dependent variable (hours of care or duration of care provided) as well as a proxy for quantifying the intensity of care (low, medium or high). We used ordered probit models to analyze the relationship between the duration of caregiving and the different types of SMA. The extended specification of the model is the following:

Caregiver’s time = β_0_ + β_1_ age + β_2_ sex + β_3_ country + β_4_ marital status + β_5_ work status+ β_6_ social services + β_7_ type of SMA.

In this sense, type I was considered as the most severe type of this disease. The sex, age of the caregiver, country, marital and work status of the caregiver and whether the patient received social services were included as control variables. Marginal effects were estimated to quantify the effect of the severity of the illness on caregiving time. Marginal effects showed the likelihood of providing more time of care due to the change of the type of SMA (the reference category was type III).

As a second purpose, the burden of care was also analyzed using the Zarit Burden Interview results as the dependent variable. This analysis was carried out using ordinary least squares (OLS) robust of heteroscedasticity. We used the same control variables as in the ordered probit models and the same main explanatory variables (type of SMA). All of the statistical analysis was carried out using the Stata/SE software (v14.2, StataCorp LLC, College Station, TX, USA).

## 3. Results

The study included data collected from 68 informal caregivers of children diagnosed with SMA, including 27 patients from Spain, 11 from the UK, 14 from Germany and 16 from France. The main characteristics of the informal caregivers and the patients used in the analysis are shown in Table 1. The median time taken by patients and caregivers to answer the 74 questions included in the survey was 34.41 min.

The patients received 10.02 (SD = 6.67) mean daily hours of care. The main caregiver provided 6.89 (SD = 4.62) mean hours per day (68% of the total time per day) vs. 3.13 mean hours (SD = 3.34) for other caregivers. Table 2 presents the duration of care provided daily by the kind of activity. BADL required 6.71 (SD = 4.78) hours per day, and “helping the patient to move” and helping the patients to carry out “basic hygiene, dressing or changing tasks” were the activities which the caregivers identified as requiring more time. According to the caregivers, IADL required 22.4% of their time, and 1.08 (SD = 1.59) mean hours of care were required for activities other than BADL or IADL. Caution is needed when interpreting the results by type of SMA when using a small sample size, and extrapolating these results to the entire population of SMA depending on type may lead to biases. This is a common problem in clinical and socioeconomic analyses carried out in the field of low-prevalence diseases, but it should not cease to be recognized here. Regarding the age of the patients, those aged lower than 10 years needed on average 9.72 caregiving hours (SD = 6.26), while those older than 10 years needed 10.64 h (SD = 7.58) on average (Appendix A). On the other hand, higher differences were observed when comparing the time of care needed by patients using non-invasive respiratory supply systems (12.39; SD = 6.44 mean daily hours of care) with patients who did not use non-invasive respiratory supply systems (8.17; SD = 6.25 mean daily hours of care) (Appendix A). Regarding the country, there were no remarkable differences between patients from France (9.32; SD = 8.44 mean daily hours of care) or Spain (9.10; SD = 6.41 mean daily hours of care). However, patients from the UK and Germany required the most time, with 12.50 and 10.65 mean daily hours of care, respectively (Appendix A). Finally, we did not find differences in the time needed depending on the patient’s gender (10.61 mean daily hours of care for female patients versus 9.37 in male patients), but we observed a slight difference in the time of care due to the caregiver gender. Male informal caregivers indicated they provided 12.92, while females provided 9.34 daily hours of care on average. This difference was present in all activities, but “feeding the patient” and “administering drugs/minor cures” were the two care activities where the differences were higher (1.67 and 0.94 mean daily hours more if the caregiver were male, respectively). However, we should take these results with caution, considering that only 19% (n = 13) of the informal caregivers were male.

The results of the statistical analysis revealed a robust relationship (*p*-values < 0.05) between the type of SMA and the burden of care. These findings were strongly demonstrated when analyzing the hours of care provided by all caregivers and the hours of care provided by the main caregiver (Table 3). In this sense, the main caregiver of a patient with type I SMA had a higher probability (39% greater) of providing more than 10 h of daily care (high intensity) than caregivers of patients with type III SMA. The analysis also showed that the main caregivers of patients with SMA type I had a lower probability (34% less) of providing fewer than five hours per day (low intensity) when compared to caregivers of patients with type III SMA. When the burden of care was assessed using the results of the Zarit Burden Interview, there was no significant association with the type of SMA (Appendix A).

## 4. Discussion

This paper reveals that a disease such as SMA, apart from its effects on the children suffering from it, is also responsible for a significant burden borne by family caregivers. The primary caregivers provided 6.89 hours per day (68% of the total time per day, taking into account the time provided by the primary informal caregiver plus other caregivers), and BADL required, on average, 6.71 h per day. These figures are in line with those obtained for other rare disorders [21]. The number of hours of informal care needed by SMA patients is very similar to the time needed by patients with other low-prevalence diseases, such as Duchenne muscular dystrophy [32], mucopolysaccharidosis [33] or juvenile idiopathic arthritis [34], and by patients with other high-prevalence diseases that cause a severe loss of personal autonomy, such as stroke [35] or Alzheimer’s disease [36,37].

Furthermore, such intensity in care tasks can result in negative effects in several dimensions of caregivers’ lives. For instance, of those informal caregivers who indicated that they were employed/self-employed, 56.1% stated they had experienced work-related problems due to the care they provided to patients. Additionally, 30.4% stated that they had to take days of absence and eight caregivers (34.8%) declared that they had difficulty in meeting their work schedules. In 2015, Quian et al. published a qualitative study that included patients with SMA, parents of individuals with SMA and clinicians who specialized in caring for patients with this rare disease [17]. They aimed to reveal the needs of patients and relatives receiving SMA care but were unable to provide as much detail about the activities as we did in our study. Apart from this single study, other authors [6,38,39,40] have written about what SMA care entails or the burden of care. However, they did not supply a definition of informal care and instead focused on whatever kind of care was needed inside or outside the household.

From the statistical analysis, we observed that the type of SMA was the main factor statistically associated with caregiving time. However, when the burden was analyzed using the Zarit score, we did not find a statistically significant association. In general terms, the Zarit Burden Interview results showed outcomes similar to those in other rare diseases, such as Duchenne muscular dystrophy [32] or mucopolysaccharidosis [33].

Some limitations of the study should be pointed out. First, it was impossible to obtain a larger data sample. This result is common in the analysis of low-prevalence diseases due to the difficulty of recruiting patients. Consequently, only eleven children with type I and fifteen children with type III SMA were included. Nevertheless, the distribution by type of SMA is similar to its prevalence since type II is the most prevalent, and the number of type II patients included in our study is relatively high in comparison with the other two SMA types. Secondly, the data collection process was carried out using a self-reported online questionnaire, which primary caregivers completed, instead of personal interviews. However, these questionnaires were developed primarily for this project, and the way they were going to be used was taken into account. Thirdly, the hours of care included in our analysis focused on BADL and IADL, and we limited the time of care to 16 h per day. Children with SMA may need attention all day, especially during intercurrent illnesses; therefore, we could be underestimating the time of care needed. Lastly, we did not collect whether patients lived in rural or urban areas. Some authors have revealed greater unmet needs in receiving support from formal services in rural areas [41]. It has also been underlined that informal caregivers from rural areas need more healthcare resources [42]. Therefore, the burden supported by caregivers might depend on where they are living, but, unfortunately, no information about this aspect could have been collected in the data.

There is a growing interest in the care of people with rare diseases throughout Europe [43]. Different stakeholders are pointing out the need to carry out thorough research in this area to reveal the real effects that such progressive, rare diseases have on society, using a broad outlook [44]. In fact, it has been suggested that new statistical approaches and study designs, in addition to randomized controlled trials (RCTs) and information from sources of big data, should be incorporated to adequately capture the effect of new healthcare technologies on rare diseases [45]. There are specific challenges to be addressed in the evaluation of health technologies in rare diseases, especially when economic aspects are included in the evaluation. The advantage of incorporating societal costs (such as that of informal care or loss of productivity) in the economic evaluation of healthcare technologies or programs has already been widely discussed [10,46]. Furthermore, some authors have begun to suggest the inclusion of future unrelated medical and non-medical costs because of the years gained in cost-effectiveness [47]. However, there is a lack of specific guidelines for performing economic assessments of rare diseases. Another important issue to be considered is the role played by professional/social services when analyzing informal caregiving provided [48,49]. Namely, determining whether formal care can compensate, reinforce, replace or complement informal care. This might help to understand the differences between results found in different countries [50]. This issue has been studied in the literature, resulting in a high observed heterogeneity depending on the disease, types of disabilities identified, and, particularly, depending on geographical location due to the cultural and organizational structure of the long-term care systems. In fact, the public budget or public economic effort designated to such services varies significantly between countries, ranging from 0.5% or 0.9% of its GDP in Portugal and Spain, respectively, to 3.5% and 3.7% in the Netherlands and Norway, respectively [51].

To date, and to the best of our knowledge, this paper is the first study that has estimated the number of hours of informal care required by people with SMA and the burden associated with family caregiving. As shown in the figures obtained from this analysis, the role played by the families of children with SMA should be considered when policy-makers design any programs or strategies focused on improving the allocation of scarce resources. A new therapeutic scenario is transforming the expectations of families and health professionals. In particular, intrathecal antisense oligonucleotide administration to modify splicing of the SMN2 gene [52] has received regulatory approval as the first drug treatment for SMA. Other recently approved approaches include intravenous gene therapy based on self-complementary adeno-associated virus with SMN1 [53] and a splicing modifier oral compound [54]. In this novel context, the prospects for the care and follow-up of these patients are changing. The evolving phenotype of SMA now needs to be considered beyond the clinical trials and the changing of the epidemiological landscape of SMA types [55]. These advances will influence the future healthcare of SMA patients, switching to a more proactive approach as opposed to the reactive measures of complications and palliative care [56]. Moreover, future policy design will be important in addressing the low level of participation of caregivers in the labor force, as will measures to achieve greater workforce flexibility, respite care and financial support for families. Further research with increased sample sizes, as well as an evaluation of the influence of new therapies on patients’ health-related quality of life (HRQoL) and the burden on caregivers (e.g., health, work-related problems and leisure activities), are needed.

## 5. Conclusions

SMA has a high social impact, not only on the patients who suffer from this disease but also on their caregivers. The severity of SMA is positively associated with the duration of informal care. The hours of informal care provided could, therefore, reflect a more understandable and quantifiable burden on families and relatives, which could be a useful tool when designing specific public policies aimed at reducing the impact of this disease in the family sphere.

## Figures and Tables

**Table 1 ijerph-17-08989-t001:** Sample characteristics by type of spinal muscular atrophy (SMA).

Characteristic/Outcome	Type I (*n* = 11)	Type II (*n* = 42)	Type III (*n* = 15)	Total (*n* = 68)
Caregivers				
Sex female, *n* (%)	8 (70)	34 (81)	13 (87)	55 (81)
Age ^a^, mean (SD)	42.1 (11.7)	39.5 (9.0)	39.3 (7.6)	39.9 (9.1)
Employed/self-employed, *n* (%)	7 (64)	24 (57)	10 (66)	41 (60)
Marital status ^a^, *n* (%)				
Single	7 (70)	6 (14)	1 (7)	14 (21)
Married or cohabiting	3 (30)	32 (76)	13 (93)	48 (73)
Divorced/separated	0 (0)	4 (10)	0 (0)	4 (6)
Zarit ^b^, mean (SD)	25.1 (23.2)	34.8 (13.3)	29.6 (17.1)	31.9 (16.5)
Patients				
Sex female, *n* (%)	9 (82)	20 (48)	7 (47)	36 (53)
Age ^c^, mean (SD)	9.8 (7.7)	7.7 (4.6)	6.6 (4.2)	7.0 (5.2)
Education ^d^, *n* (%)				
Educated at an ordinary school	7 (64)	3 (8)	7 (46)	17 (26)
Educated at an ordinary centre with special support	2 (18)	22 (56)	6 (40)	30 (46)
Educated at a special needs education centre	0 (0)	3 (8)	0 (0)	3 (5)
Home schooled	0 (0)	1 (3)	1 (7)	2 (3)
Nursery school	0 (0)	8 (21)	1 (7)	9 (14)
Not receiving education	2 (18)	2 (5)	0 (0)	4 (6)
Genetic test confirmation of SMA (yes), *n* %	9 (82)	35 (83)	12 (80)	56 (82)

^a^ missing values = 2; ^b^ missing values = 13; ^c^ missing values = 6; ^d^ missing values = 3.

**Table 2 ijerph-17-08989-t002:** Mean hours (SD) of informal caregiving provided daily to patients with SMA by type of SMA.

Activity	Type I		Type II		Type III		All	
	Main Caregiver	All Caregivers	Main Caregiver	All Caregivers	Main Caregiver	All Caregivers	Main Caregiver	All Caregivers
Basic hygiene, dressing	1.82 (1.01)	3.23 (2.18)	1.26 (1.11)	1.62 (1.2)	0.58 (0.36)	0.91 (0.58)	1.2 (1.04)	1.73 (1.48)
Feeding the patient	1.61 (1.88)	2.5 (2.67)	1.33 (1.45)	1.71 (1.92)	0.45 (0.63)	0.69 (1.09)	1.18 (1.43)	1.61 (1.97)
Bathing or showering	0.86 (0.53)	1.61 (1.16)	0.7 (0.6)	1 (0.96)	0.38 (0.36)	0.71 (0.64)	0.66 (0.56)	1.03 (0.96)
Helping the patient to move	1.18 (0.9)	2.05 (1.21)	2.15 (2.38)	2.75 (2.71)	0.74 (1.32)	1.39 (2.61)	1.68 (2.08)	2.34 (2.54)
*Total BADL*	*5.47 (2.71)*	*9.38 (4.77)*	*5.44 (3.71)*	*7.08 (4.42)*	*2.15 (2.34)*	*3.71 (4.52)*	*4.72 (3.54)*	*6.71 (4.78)*
Cooking and preparing special meals	0.82 (0.72)	1.55 (1.27)	0.65 (0.82)	0.91 (1.12)	0.18 (0.36)	0.25 (0.49)	0.57 (0.75)	0.87 (1.1)
Administering drugs/minor cures	1.18 (0.84)	1.98 (1.58)	1.07 (1.19)	1.35 (1.47)	0.64 (1.31)	0.83 (1.53)	0.99 (1.17)	1.33 (1.52)
Other IADL ^a^	0.03 (0.1)	0.03 (0.1)	0.02 (0.14)	0.05 (0.3)	0 (0)	0 (0)	0.02 (0.12)	0.03 (0.24)
*Total IADL*	*2.03 (1.12)*	*3.55 (2.17)*	*1.74 (1.59)*	*2.31 (2.18)*	*0.83 (1.3)*	*1.08 (1.51)*	*1.59 (1.5)*	*2.24 (2.16)*
Other activities directly related to the disease	0.8 (0.73)	1.39 (1.25)	0.68 (1.12)	1.09 (1.72)	0.17 (0.36)	0.8 (1.49)	0.59 (0.96)	1.08 (1.59)
*All activities*	*8.3 (3.6)*	*14.32 (6.94)*	*7.86 (4.8)*	*10.48 (6.41)*	*3.14 (2.42)*	*5.59 (4.73)*	*6.89 (4.62)*	*10.02 (6.68)*

Note: time of informal care was capped at a maximum of 16 h per day per caregiver. ^a^ Other IADL included the time spent on medical visits, diagnostic tests, traveling, financial, administrative or legal affairs and social and leisure activities.

**Table 3 ijerph-17-08989-t003:** Results of the statistical analysis (marginal effects) of the time of informal care of patients with SMA.

**Informal Caregiving Time ^a^ (All Time of Care, Main Caregiver Plus Other Caregivers)**			
	**High Burden of Care (more than 10 h per day)**	**Intermediate Burden of Care (between 5 and 10 h per day)**	**Low Burden of Care (less than 5 h per day)**
	**dy/dx (Standard Error)**	**dy/dx (Standard Error)**	**dy/dx (Standard Error)**
Type of SMA (reference category = type III)			
Type I	0.36 * (0.17)	−0.05 (0.04)	−0.31 * (0.15)
Type II	0.30 ** (0.10)	−0.04 (0.04)	−0.26 ** (0.08)
*n*			64
LR chi^2^			35.9
Pseudo R^2^			0.26
**Informal caregiving time ^a^ (main caregiver’s time)**			
	**dy/dx (Standard Error)**	**dy/dx (Standard Error)**	**dy/dx (Standard Error)**
Type of SMA (reference category = type III)			
Type I	0.39 * (0.20)	−0.05 (0.05)	−0.34 * (0.17)
Type II	0.44 ** (0.12)	−0.06 (0.06)	−0.38 ** (0.09)
*n*			64
LR chi^2^			20.87
Pseudo R^2^			0.15

* *p* < 0.05 ** *p* < 0.01. Note: regression models include sex, age, country, marital and job status of caregivers and whether the patient with SMA is receiving social services due to the illness as control variables. ^a^ These values represent the marginal effects obtained using ordered probit models.

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
