# Peer review of "The Burden of Spinal Muscular Atrophy on Informal Caregivers"

_ijerph, 2020, doi:10.3390/ijerph17238989_

Round 1
Reviewer 1 Report
Isaac Aranda-Reneo et al. described the burden on the primary caregivers for children with spinal muscular atrophy in Spain, Germany, the United Kingdom and France. According to them, the disease severity was associated with more time of care and a higher burden on the caregivers. However, Zarit score did not show statistically significant association with the disease severity. Finally, they suggested the possibility that new drugs for SMA will influence the future healthcare of SMA patients. The study results are convincing, and the authors’ logic is clear. The contents of this paper may be a basis of the policy design for SMA families.
[Minor issues]
(1) Line 186: Here, “ho per day” may be “hours per day”.
(2) Lines 193-195: The sentence of “Age of the patients seem to be slightly higher in patients aged lower than 10 years old (9.72; SD=6.26 mean daily hours of care) in comparison with patients aged higher than 10 years (10.64; SD=7.58 mean daily hours of care)” may have typing errors.
Reviewer 2 Report
This study investigated the burden of care for formal and informal caregivers of individuals with spinal muscular atrophy. This is an important area of research. Please see suggestions for revision below
Line 77: I suggest deleting the word “an”
Line 81: Briefly explain what is meant by “overt total costs”
Line 81: At the end of this line, change “costs” to “cost”
Line 82: “the 77% of total cost-of-illness” – delete the word “the”. Insert the word “and” before “80% in France”
Line 101: Please provide the name of the ethics committee that approved the study and provide a clear statement about informed consent from participants.
Line 114: I suggest deleting the words “In fact”
Is your sample size of 68 adequate for the number of variables in the model for caregiver’s time (line 163)? Please provide justification for the sample size with respect to the model.
Line 186: Change “ho” to “hours”
Lines 193-195: “Age of the patients seem to be slightly higher in patients aged lower than 10 years old (9.72; SD=6.26 mean daily hours of care) in comparison with patients aged higher than 10 years (10.64; SD=7.58 mean daily hours of care)” – this sentence is unclear and I am not sure I understand what you are saying here. Please re-word this sentence.
Line 196: “On the other hand, it was observed higher differences” – re-word to “On the other hand, higher differences were observed…”
Line 201: “who most time required” – change to “who required most time”
Line 249: Change “we did not find statistically association” to “we did not find a statistically significant association”
Lines 278-280: “Namely, determining whether formal care compensate, reinforce, replace or complement informal care might help to understand the differences results found across countries[48].” – change to “Namely, determining whether formal care compensates, reinforces, replaces or complements informal care. This might help to understand the different results found across countries[48].”
Did you assess whether families were living in urban or rural areas? Does the burden of care differ for these families?
Reviewer 3 Report
This is a full-length research report of identifying non-professional burden, especially the duration of time for care of patients suffering from spinal muscular atrophy (SMA), and comparisons of the burden between types of SMA in order to reveal the severity of types of SMA in terms of non-professional human burden in line with daily life. The authors have tried to compare the factors as indexed by scores via some reliable measures of questionnaire towards informal caregivers. Based on data from the questionnaire, the authors revealed a significant association of time of care with the severity of the disease, especially type I vs. type III of SMA. As the authors described in the manuscript despite the limitations of the study, the purpose of the present study as well as the results is believed to contribute the re-consideration of professional healthcare system(s) and their economic burden. This issue has progressively been focused worldwide, and overall impact of their research seems to be considered strong.
I have following comments/suggestions to consider the revision:
- To enhance the strength of the manuscript, the authors might add to the manuscript their perspective for care time differences among the severity of SMA. Most importantly the readers wish to know the perspective as a take-home message of the present study.
- Is there any difference between the sex of the patients with SMA in terms of care time provided by informal caregivers? Differences of care time in terms of sex seem to be a predominant factor to be considered.
- Please identify the exact name of Ethics Committee that had approved this study and also the approved number(s) assigned to this study (page 3, line 101).
Round 2
Reviewer 3 Report
The authors have done a good job responding to reviewer comments and concerns in their revision. I believe the manuscript is significantly improved as a result.